# SARS-CoV-2 nucleocapsid protein phase-separates with RNA and with human hnRNPs

Theodora Myrto Perdikari[1,†], Anastasia C Murthy[2,†], Veronica H Ryan[3,†], Scott Watters[4,†], Mandar T Naik[4] & Nicolas L Fawzi[4,5,*] ID

## Abstract

**Tightly packed complexes of nucleocapsid protein and genomic RNA form the core of viruses and assemble within viral factories, dynamic compartments formed within the host cells associated with human stress granules. Here, we test the possibility that the multivalent RNA-binding nucleocapsid protein (N) from severe acute respiratory syndrome coronavirus 2 (SARS-CoV-2) condenses with RNA via liquid–liquid phase separation (LLPS) and that N protein can be recruited in phase-separated forms of human RNA-binding proteins associated with SG formation. Robust LLPS with RNA requires two intrinsically disordered regions (IDRs), the N-terminal IDR and central-linker IDR, as well as the folded C-terminal oligomerization domain, while the folded N-terminal domain and the C-terminal IDR are not required. N protein phase separation is induced by addition of non-specific RNA. In addition, N partitions *in vitro* into phase-separated forms of full-length human hnRNPs (TDP-43, FUS, hnRNPA2) and their low-complexity domains (LCs). These results provide a potential mechanism for the role of N in SARS-CoV-2 viral genome packing and in host-protein co-opting necessary for viral replication and infectivity.**

**Keywords** biomolecular condensates; heterogeneous nuclear ribonucleoproteins; intrinsically disordered proteins; liquid-liquid phase separation; RNA-binding proteins

**Subject Categories** Microbiology, Virology & Host Pathogen Interaction; RNA Biology

**The EMBO Journal (2020) 39: e106478**

## Introduction

The spread of the highly infectious severe acute respiratory syndrome coronavirus 2 (SARS-CoV-2) is responsible for the ongoing global pandemic of Coronavirus Disease 2019 (COVID-19) (Zhu *et al*, 2020). The novel SARS-CoV-2 is an enveloped, non-segmented, positive-sense, single stranded ~ 30 kb RNA virus of the family *Coronaviridae* (Zhou *et al*, 2012). This family includes the related SARS-CoV (SARS) (Drosten *et al*, 2003) and Middle East respiratory syndrome (MERS) (Zaki *et al*, 2012) coronaviruses, which have both caused previous outbreaks of pneumonia. Like all coronaviruses, SARS-CoV-2 forms a virion including its genomic RNA (gRNA) packaged in a particle comprised of four structural proteins—the crown-like spike (S) glycoprotein that binds to human ACE2 receptor to mediate the entry of the virus in the host cell (Walls *et al*, 2020; Wang *et al*, 2020), the membrane (M) protein that facilitates viral assembly in the endoplasmic reticulum, the ion channel envelope (E) protein, and the nucleocapsid protein (N) that assembles with viral RNA to form a helical ribonucleoprotein (RNP) complex called the nucleocapsid (Masters, 2006; Singh Saikatendu *et al*, 2007). Though many current therapeutic efforts have focused on disrupting viral attachment to host cells (Du *et al*, 2009) and preventing viral protease function (Zhang *et al*, 2020), the molecular mechanisms that underlie the assembly of the SARS-CoV-2 nucleocapsid through the binding of nucleoprotein to RNA are poorly understood and therefore have remained an uninvestigated target to inhibit viral replication.

Nucleocapsid protein of SARS-CoV-2 is a multidomain 46 kDa RNA-binding protein which is predicted to have 40% of its primary sequence remaining intrinsically disordered in addition to the two known folded domains (Fig 1A). N has a folded N-terminal domain (NTD) that participates in RNA-binding preceded by a 44-amino acid N-terminal disordered region ($N_{IDR}$) and followed by a 73-amino acid linker ($linker_{IDR}$) whose SR dipeptides are phosphorylated in cells infected by SARS-CoV-2 (Bouhaddou *et al*, 2020) (Fig 1A). The flexible linker is followed by the folded C-terminal dimerization domain (CTD) and a 52-amino C-terminal disordered tail region ($C_{IDR}$; Fig 1B). Previous studies of SARS-CoV N (91% sequence identity with SARS-CoV-2 N) have shown that the NTD, the CTD, and disordered regions can bind RNA cooperatively to promote RNP packaging (Chang *et al*, 2009) and chaperoning (Zúñiga *et al*, 2006). The structural details of these protein-RNA

1 Center for Biomedical Engineering, Brown University, Providence, RI, USA
2 Molecular Biology, Cell Biology & Biochemistry Graduate Program, Brown University, Providence, RI, USA
3 Neuroscience Graduate Program, Brown University, Providence, RI, USA
4 Department of Molecular Pharmacology, Physiology, and Biotechnology, Brown University, Providence, RI, USA
5 Robert J. and Nancy D. Carney Institute for Brain Science, Brown University, Providence, RI, USA
*Correspondence: Nicolas L. Fawzi (nicolas_fawzi@brown.edu)
†These authors contributed equally to this work

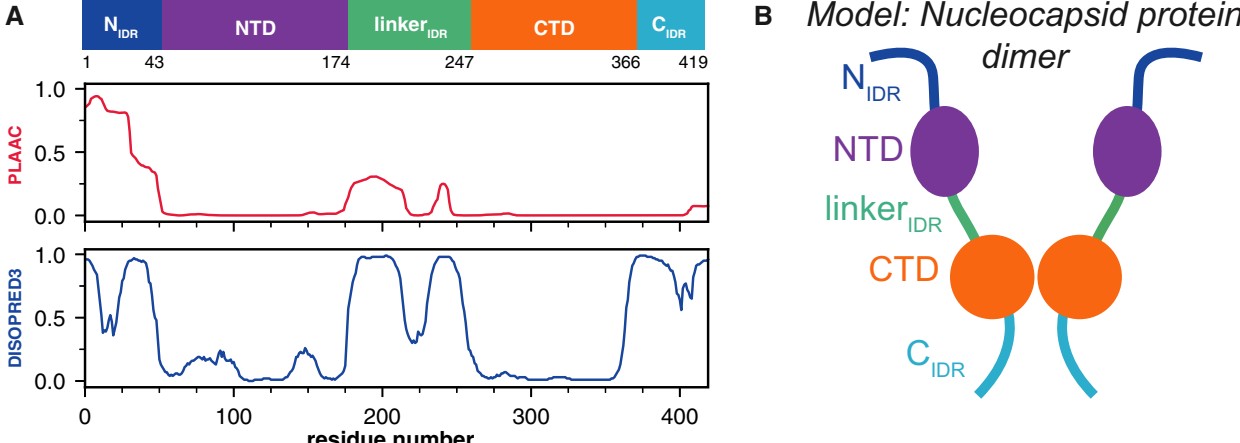

**Figure 1. Domain structure and sequence features of SARS-CoV-2 nucleocapsid protein.**

A  Domain structure (top), prion-like sequence propensity (PLAAC, middle), and predicted disorder propensity (DISOPRED3) vs. sequence of SARS-CoV-2 N. N contains three putatively disordered regions, a globular N-terminal and a globular C-terminal oligomerization domain.

B  Schematic of the folded and disordered domains of a SARS-CoV-2 N dimer.

interactions are beginning to come into focus. A recent solution NMR structure of SARS-CoV-2 N NTD-RNA complex suggested a right hand-like fold and highlights the role of arginine motifs and electrostatic interactions (preprint: Dinesh *et al*, 2020). Furthermore, superimposition of the unbound state with previously solved coronavirus nucleoproteins bound to RNA revealed some potential protein-RNA recognition interactions involving Arg89, Tyr110, and Tyr112 in nitrogenous base binding (Kang *et al*, 2020). The CTD dimerization domain of SARS-CoV has been suggested to organize into an octamer stabilized via electrostatic interactions enhanced by phosphorylation (Chang *et al*, 2013) to promote superhelical packaging of viral RNA (Chen *et al*, 2007). Some specifics of the NTD of other viral nucleocapsid proteins in viral genome packaging has been investigated in HCoV-OC43 N-NTD (Grossoehme *et al*, 2009) and mouse hepatitis virus (MHV) N-NTD which organizes gRNA via specific interactions with a packaging signal (PS) located 20.3 kb from the 5′ end of gRNA (Kuo *et al*, 2016).

In recent years, liquid–liquid phase separation (LLPS) has emerged as a common cellular process to organize biological material into compartments. Many of these biomolecular condensates assembled by LLPS are multicomponent condensates composed of multivalent RNA-binding proteins containing intrinsically disordered regions (IDRs) and RNA (Banani *et al*, 2017). Moreover, RNA can also self-assemble *in vitro* without requiring protein components via intermolecular interactions such as Watson-Crick base pairing and Hoogsteen base pairs typical of G-quadruplexes (Van Treeck *et al*, 2018) while accumulation of nucleotide repeats associated with neurodegenerative diseases such as "GGGGCC" can induce RNA gelation (Freibaum *et al*, 2015; Jain & Vale, 2017). Many biomolecular condensates are thought to be stabilized by weak, multivalent protein–protein, and protein–RNA interactions and sequester and concentrate proteins involved in RNA processing, stress response and gene silencing (Alberti & Carra, 2018). In eukaryotes, histone proteins are known to promote the compaction of chromatin into nuclear condensates (Gibson *et al*, 2019). Bacterial nucleoprotein complexes also have been shown to organize genomic DNA via phase separation (Monterroso *et al*, 2019), suggesting that LLPS may serve to organize genome packaging across the domains of life. Recent evidence suggests that similar higher order genome organization is also present in viruses. For example, the measles virus nucleoprotein (MeV N) assembles with genomic RNA into a rigid helical capsid (Milles *et al*, 2016) which also undergoes LLPS in the presence of the phosphoprotein (P) (Guseva *et al*, 2020). Like the eukaryotic heterochromatin protein 1 (HP1) which bridges chromatin regions and can undergo LLPS (Larson *et al*, 2017), SARS-CoV-2 N is oligomeric with multiple binding sites for the genomic nucleic acid separated by disordered linkers. Furthermore, both the $N_{IDR}$ and $linker_{IDR}$ of SARS-CoV-2 N have a "prion-like" sequence composition (defined by resemblance in amino acid composition to the polar-residue-rich domains of yeast prion proteins) (King *et al*, 2012) that are known to contribute to phase separation of other RNA-binding proteins (Cascarina & Ross, 2020) (Fig 1A). Hence, it is important to understand if SARS-CoV-2 N could also use phase separation to condense its genome.

Viruses hijack the host cell environment to facilitate gRNA transport from the site of viral genome replication to the site of viral assembly and maximize the replication efficiency by disrupting the organization of cellular organelles (Novoa *et al*, 2005; Onomoto *et al*, 2014). A recent study on the network of protein–protein contacts formed by SARS-CoV-2 structural proteins shows that N interacts with human ribonucleoproteins known to be involved in the formation of phase-separated protein-RNA granules (Gordon *et al*, 2020). Several of these proteins such as G3BP1/2 contain disordered regions with prion-like sequence characteristics known to contribute to the formation of stress granules (SGs) (Guillén-Boixet *et al*, 2020; Yang *et al*, 2020), membraneless organelles that store translationally silent mRNA when the cell is exposed to stress to regulate mRNA metabolism. Numerous studies have shown viral

invasion can interfere with SG formation (White & Lloyd, 2012) via inhibition of post-translational modifications (Linero *et al*, 2011), exclusion of SG components such as TIA-1 and G3BP (Emara & Brinton, 2007; Nikolic *et al*, 2016), and formation of stable viral RNP complexes with essential SG proteins (Abrahamyan *et al*, 2010). Hence, testing if and how SARS-CoV-2 N can enter phase-separated assemblies formed by other ubiquitous SG human proteins may serve as a model for N interactions with SG proteins in cells.

To provide biochemical insight into the mechanisms by which SARS-CoV-2 nucleocapsids may assemble and co-opt host proteins, here we test if SARS-CoV-2 N can undergo phase separation *in vitro* in an RNA-dependent manner, which domains contribute to N phase separation, and if N is able to partition into phase-separated droplets formed by intact human ribonucleoproteins or their disordered domains.

## Results

### N forms higher order oligomers

The SARS-CoV nucleocapsid protein contains multiple regions implicated in self-interactions (Luo *et al*, 2005; Yu *et al*, 2005; Chang *et al*, 2013; Cong *et al*, 2017), so we first sought to determine if the SARS-CoV-2 nucleocapsid protein (N) was capable of higher order assembly. We purified recombinant full-length SARS-CoV-2 N with a TEV protease cleavable N-terminal maltose binding protein (MBP) tag to enhance solubility, due to previous studies showing that the N-RNA complex was largely insoluble (Chang *et al*, 2009). Even with the solubility tag, some of the N protein deposited into inclusion bodies during bacterial expression. However, purification of MBP-N from the soluble fraction was efficient at high salt concentration via standard immobilized metal affinity chromatography. The chromatogram from subsequent preparative-scale gel filtration chromatography of both MBP-tagged and cleaved (tag removed) N have major absorbance peaks eluting much earlier than would be expected of their monomeric species (90 kDa expected at about 230 ml, and 46 kDa expected at about 240 ml, respectively; Fig 2A). When analyzed by SDS–PAGE, the fractions containing these peaks correspond to the MBP-N and cleaved N, respectively, with some minor degradation products but no species of greater molecular weight (Fig 2B and C). These data imply that N forms stable but soluble multimers.

### N undergoes LLPS *in vitro*

The SARS-CoV nucleocapsid protein associates with viral genomic RNA to form a ribonucleoparticle and has segments enriched in polar residues and serine/arginine motifs (Castello *et al*, 2012; Chong *et al*, 2018), characteristics of some proteins that can undergo LLPS. Therefore, we hypothesized that SARS-CoV-2 N is able to form liquid-like compartments to sequester RNA. To test if N is able to phase separate, we adapted our established assay used for studies of LLPS-prone proteins where the solubilizing and phase separation inhibiting MBP tag (Boehning *et al*, 2018) is liberated from the phase separation prone protein by addition of TEV protease to induce LLPS (Burke *et al*, 2015). (Importantly, MBP is

not prone to self-interaction nor does it interact with RNA (Wang *et al*, 2015)). Briefly, we prepared samples containing MBP-tagged full-length N, added TEV, and tested for phase separation by measuring the resulting turbidity of the solution and confirmed phase separation via microscopy. We performed the experiment at a variety of solution conditions to test for basic biochemical features of the interactions that mediate phase separation. First, we probed the RNA-dependence of N phase separation by testing whether N could undergo LLPS in the presence and absence of torula yeast RNA extract (buffer exchanged by spin desalting column to remove ions and small RNA pieces; Fig 3). Like in our previous work examining the impact of RNA binding on phase separation of FUS which binds many RNA sequences and structures (Schwartz *et al*, 2013), here we used RNA extract as N has also been shown to bind with little specificity to nucleic acids including ssRNA, ssDNA, and dsDNA (Yu *et al*, 2005; Chang *et al*, 2009). At pH 7.4, mixtures containing 50 μM MBP-N with 0.3 mg/ml RNA in the presence of TEV protease displays initial increased turbidity followed by a decrease over time (Fig 3A), characteristic for the formation of turbid liquid droplet assemblies triggered by MBP cleavage which then fuse and settle (Monahan *et al*, 2017; Ryan *et al*, 2018). This increase in turbidity is coupled with the appearance of small, spherical droplets visible by microscopy that are not found in the absence of RNA (Fig 3B). Hence, LLPS of N is enhanced by addition of RNA and release from the MBP fusion. Interestingly, at lower pH conditions, irregularly shaped assemblies are observed in the absence of RNA at these conditions (Fig 3C, left), suggesting protein–protein interactions are enhanced at low pH, though the molecular origin of this effect is not yet clear. The fact that no phase separation is observed in the absence of TEV cleavage suggests that interactions at these conditions are not mediated by artifactual contacts between MBP and N, as these should be enhanced when the proteins are fused. Furthermore, addition of RNA appears to make the phase-separated condensates more spherical (Fig 3C, right), suggesting that RNA binding may "fluidize" N, as observed for other RNA-protein condensates (Elbaum-Garfinkle *et al*, 2015). Still, the turbidity of these low pH samples increase and persists over time as has been observed for protein aggregates or gels (Murthy *et al*, 2019) (Fig 3A). Consistent with this hypothesis, at lower pH conditions the condensates are not spherical and show signs of incomplete fusion even in the presence of RNA (Appendix Fig S1). Given that RNA structure and folding can be modulated by multivalent cations especially magnesium (Stein & Crothers, 1976; Bowman *et al*, 2012), we also tested for the effect of addition of $MgCl_2$ or $CaCl_2$ on LLPS of N. We find that high amounts (2 mM) of magnesium or calcium do not substantially alter LLPS of N (Appendix Fig S2). Together, these data suggest that SARS-CoV-2 N is able to undergo LLPS with RNA in buffer solutions mimicking physiological conditions.

### N LLPS is modulated by RNA concentration and ionic strength

Next, we sought to probe the biochemical details of how protein–protein and protein–RNA interactions contribute to N phase separation (Fig 4). To this end, we conducted turbidity and microscopy experiments with fixed protein concentration and increasing RNA (and therefore varying protein:RNA mass ratios) in low salt conditions (Fig 4A and C). First, in low salt conditions in the absence of

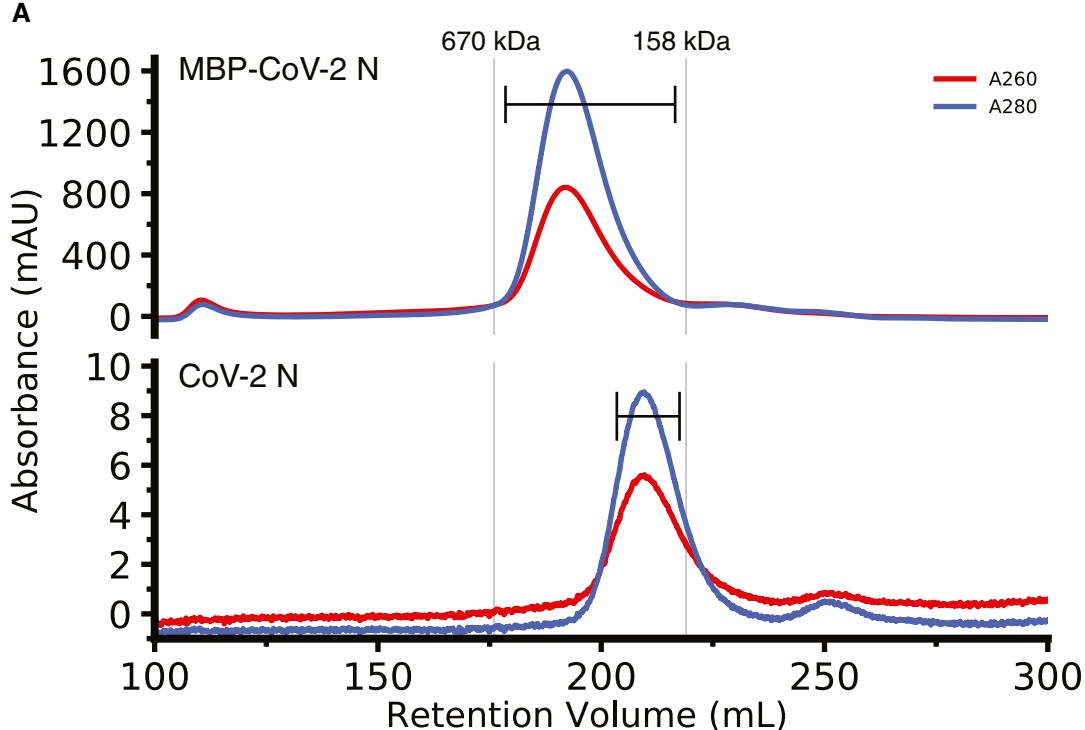

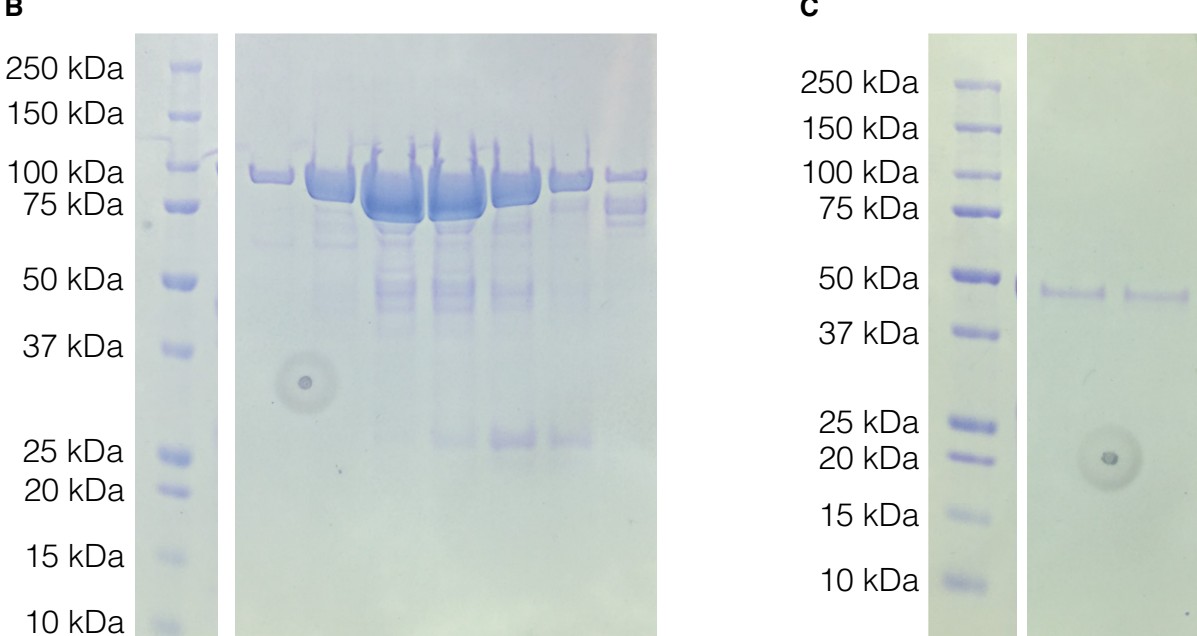

**Figure 2. MBP-CoV-2 N and cleaved CoV-2 N elute larger than their predicted molecular weights.**

A  Superdex 200 26/600 gel filtration chromatogram of ~ 7ml of ~ 500 μM MBP-tagged SARS-CoV-2 N (top) and ~ 300 μl ~ 50 μM SARS-CoV-2 N (previously cleaved from MBP) (bottom). Vertical lines represent peak elution volumes of gel filtration protein calibration standards. Brackets represent range of gels lanes. Predicted molecular weight of the tagged and untagged N are 90 kDa and 46 kDa, respectively, much smaller than their corresponding calibrated peak elution volumes, consistent with oligomerization.

B  SDS–PAGE showing the peak fractions of the MBP-SARS-CoV-2 N chromatogram showing expected 90 kDa molecular weight.

C  SDS–PAGE showing the peak fractions of the cleaved SARS-CoV-2 N chromatogram showing expected 46 kDa molecular weight. Circular feature in SDS–PAGE gels is a divet in plastic on which pictures were taken.

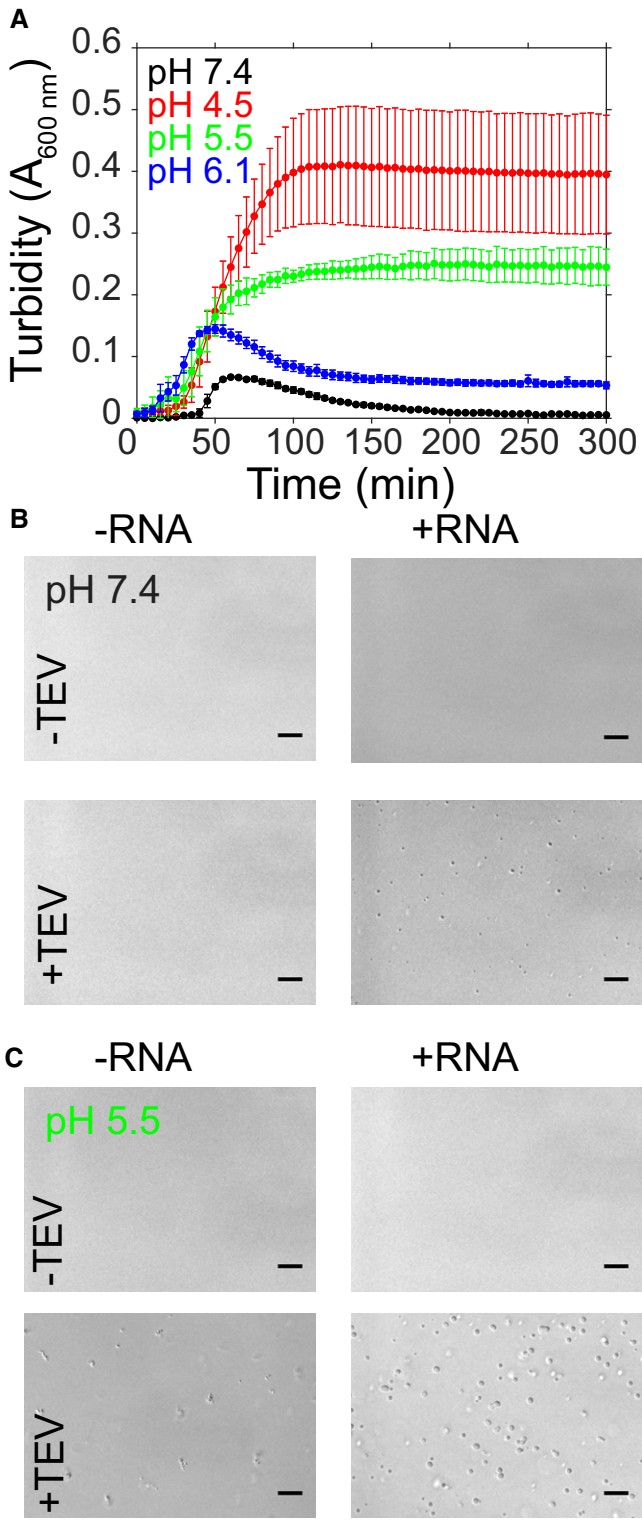

**A**

**B** -RNA  +RNA
pH 7.4
-TEV
+TEV

**C** -RNA  +RNA
pH 5.5
-TEV
+TEV

**Figure 3. SARS-CoV-2 nucleocapsid protein undergoes LLPS at physiological conditions.**

A–C (A) Phase separation over time as monitored by turbidity of 50 μM MBP-N after addition of TEV protease in the presence of salt (183 mM NaCl) and 0.3 mg/ml RNA at pH 7.4 (50 mM Tris), pH 6.1, pH 5.5, or pH 4.9 (pH 6.1 and below in 20 mM MES). Error bars represent standard deviation of three replicates. For samples at (B) pH 7.4 or (C) pH 5.5, DIC micrographs of 50 μM MBP-N in the presence of salt (183 mM NaCl) without or with TEV protease (to cleave MBP from N) and without or with 0.3 mg/ml desalted total torula yeast RNA. Scale bars represent 50 μm.

(Nott *et al*, 2015) and anticipated for the disordered regions of N (Moosa & Banerjee, 2020). At conditions of 1:0.25 MBP-N:RNA, the turbidity of the solution and the apparent number and size of the liquid droplets are enhanced compared to without RNA (Fig 4A and C). Interestingly, at higher RNA concentrations turbidity and droplet formation are diminished (Fig 4A and C), a characteristic of reentrant phase transition (Banerjee *et al*, 2017). To test if the electrostatic interactions between N and RNA are important for phase separation, we measured turbidity accompanied by microscopy of solutions containing varying sodium chloride concentrations (Fig 4B and D). We found that at higher salt concentrations (300 mM and 1M), both turbidity and droplet formation was reduced. Together, these data show protein–RNA interactions stimulate N phase separation and suggest that screening the electrostatic interactions between N and RNA reduces LLPS.

### The N-terminal and linker intrinsically disordered regions and the C-terminal dimerization domain are essential for robust N LLPS

To probe the role of each domain in RNA-mediated LLPS of N, we designed and purified a series of MBP-N variants deleting one domain/region at a time (Appendix Fig S3). N contains two structured domains: the N-terminal domain (NTD, aa: 44–174) and the C-terminal dimerization domain (CTD, aa: 248–366), both of which bind nucleic acids (preprint: Dinesh *et al*, 2020; Kang *et al*, 2020). After cleavage of the solubilizing MBP fusion where full-length N robustly phase separates (50 μM protein, 0.5 mg/ml RNA, 70 mM NaCl), we observed that deletion of the N-terminal domain had little effect on LLPS (Fig 5A). In the absence of RNA, the turbidity was in fact slightly higher for the NTD deletion compared to that for the full-length, suggesting deletion of the NTD enhances protein–protein interactions (Fig 5B), possibly by creating a contiguous stretch of prion-like disordered sequence made up of the $N_{IDR}$ and linker$_{IDR}$. In the presence of RNA, the turbidity for the deletion of NTD is comparable to full-length (Fig 5C), suggesting that NTD-RNA contacts are not required for phase separation enhancement by RNA. In contrast to deletion of the NTD, deletion of the CTD nearly abolished formation of liquid droplets and turbidity (Fig 5A–C), consistent with the role of this domain in forming a homodimer that increases interaction valency as well as binds RNA (Chen *et al*, 2020). Regarding the disordered regions, we found that deletion of the $N_{IDR}$ modestly enhanced droplet formation and turbidity in the absence of RNA (Fig 5A and B), while, in the presence of RNA, phase separation is strongly decreased (Fig 5A and C). Similar to deletion of the folded CTD, deletion of the linker$_{IDR}$ dramatically reduces formation of droplets and turbidity associated with LLPS in

RNA, after cleavage from the MBP fusion there is an increase in turbidity along with the formation of small, spherical droplets, demonstrating that N is able to undergo LLPS in the absence of RNA (Fig 4A and C), consistent with enhanced protein-protein electrostatic or pi interactions shown for other phase-separating proteins

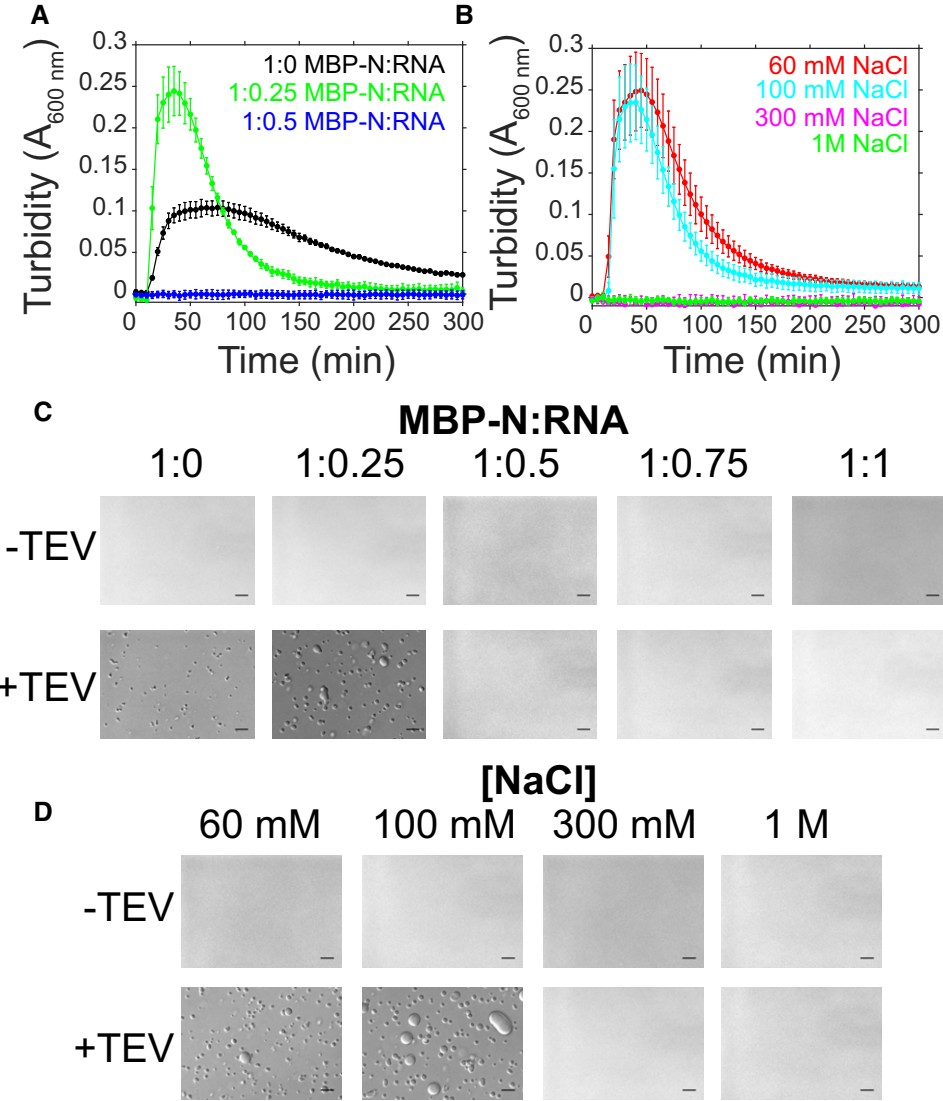

**Figure 4. SARS-CoV-2 N LLPS is modulated by salt and RNA.**

A–D Phase separation over time as monitored by turbidity of 50 μM MBP-N in 50 mM Tris pH 7.4 after addition of TEV protease (A) with varying torula yeast RNA (at 100 mM NaCl) or (B) varying NaCl concentrations (at constant RNA concentration). Error bars represent standard deviation of three replicates. DIC micrographs of 50 μM MBP-N in 50 mM Tris pH 7.4 (C) with varying torula yeast RNA concentrations (at 100 mM sodium chloride) and (D) varying sodium chloride concentrations (at constant RNA concentration), with or without TEV protease (to cleave MBP from N). Scale bars represent 50 μm.

both conditions (Fig 5A–C). Interestingly, deletion of the $C_{IDR}$ has the opposite effect, enhancing phase separation in both conditions. These data suggest the $C_{IDR}$ forms contacts or takes on conformations that are more energetically favorable in the dispersed phase (outside the droplet) than within the condensed phase (inside the droplet). In summary, these data suggest that in the presence of RNA, the $N_{IDR}$, the $linker_{IDR}$, and the folded CTD form protein–protein and/or protein–RNA contacts important for the multivalent interactions stabilizing *in vitro* LLPS of N.

**RNA sequence specificity is not necessary for enhancing N LLPS**

Having demonstrated that N is capable of LLPS in the presence of torula yeast (cellular extract) RNA, we wondered if

homopolymeric RNAs (polyA, polyC, polyG, polyU) that lack RNA sequence-specific structures would also be effective at inducing LLPS. Additionally, we wondered if any homopolymeric RNA would be more effective than another at stimulating LLPS. To compare the effect of homopolymeric RNAs on nucleocapsid phase separation, we first performed turbidity measurements at conditions where N phase separates alone. Importantly, the homopolymeric RNA stock solutions are all buffer exchanged via a spin desalting column and therefore are soluble in these conditions alone. Like torula yeast RNA, all homopolymeric RNAs enhanced the turbidity of solutions of N (Fig 6A), suggesting that particular sequences or RNA structures are not essential for interaction with N. However, not all homopolymeric RNAs enhance turbidity to the same degree. We found that while addition of polyC and polyU

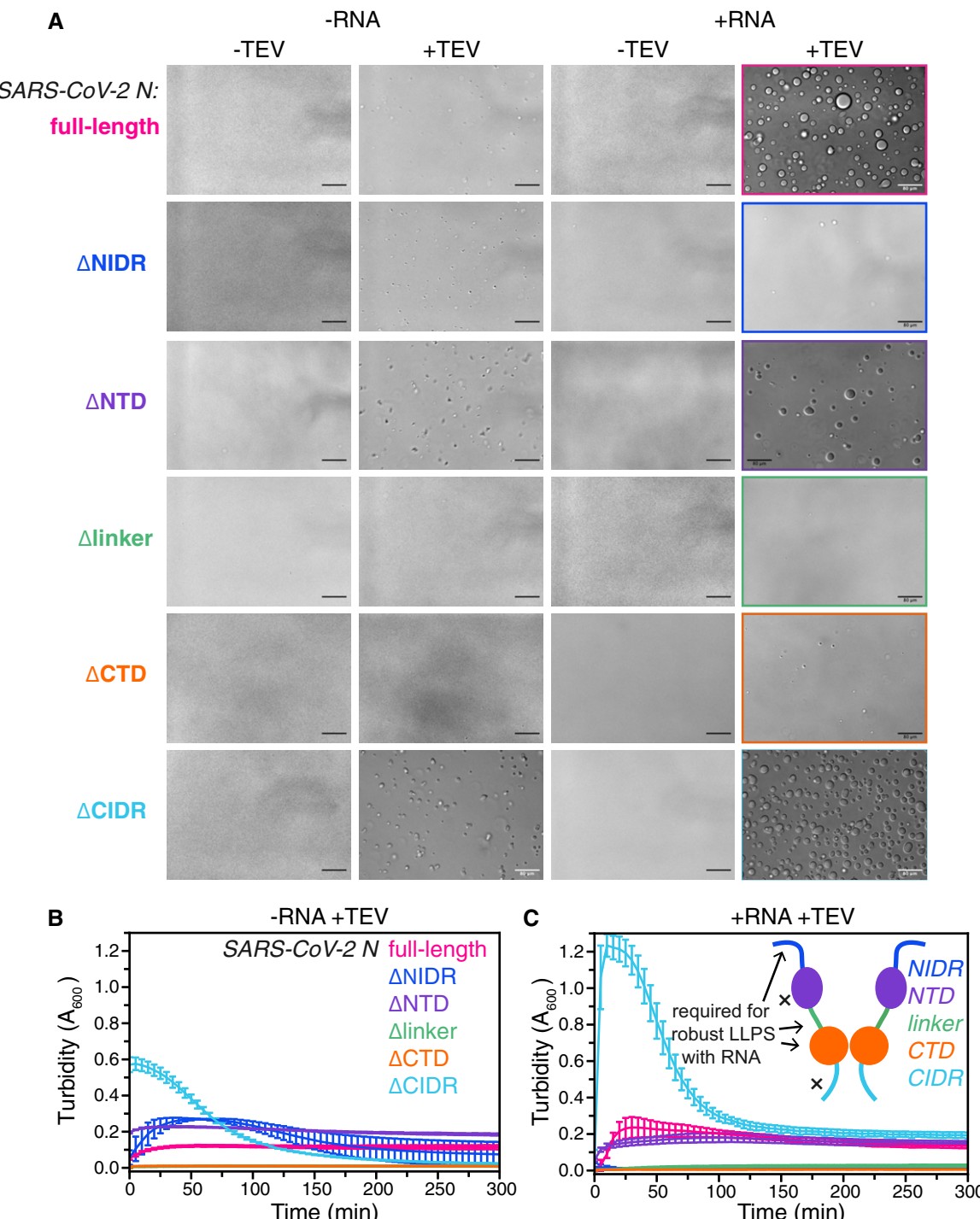

**Figure 5. The N-terminal and linker intrinsically disordered domains and the C-terminal dimerization domain are essential for robust N LLPS *in vitro*.**

A–C   DIC micrographs of 50 µM N and domain deletion variants in 50 mM Tris 70 mM NaCl pH 7.4 without and with TEV protease (to cleave MBP from N) and without or with 0.5 mg/ml desalted total torula yeast RNA. Scale bars represent 80 µm. Phase separation over time as monitored by turbidity of 50 µM full-length N or deletion variants after addition of TEV protease in the absence (B) or presence (C) of 0.5 mg/ml desalted total torula yeast RNA. Error bars represent standard deviation of three replicates.

RNAs had similar effects on N solution turbidity as addition of torula yeast RNA, polyA and in particular polyG had the ability to induce substantially more turbidity in the presence of N (Fig 6A).

However, polyG RNA drastically and quickly increases turbidity of N, and this turbidity persists over the course of the experiment, consistent with protein aggregation (Fig 6A) (Conicella *et al*, 2016)

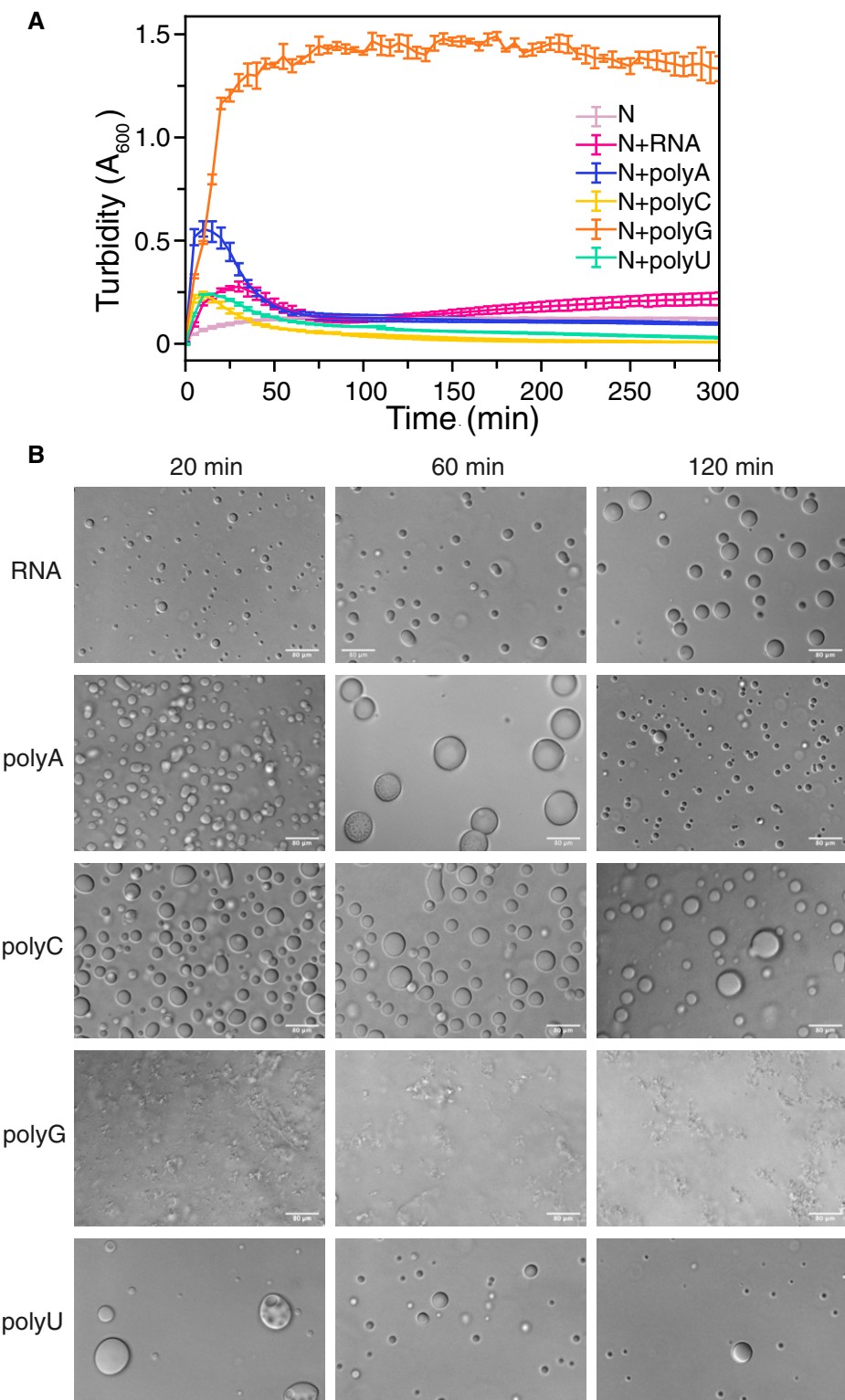

**Figure 6. SARS-CoV-2 N phase separation enhanced by non-specific RNA binding.**

A   Turbidity of N is increased in the presence of torula yeast RNA and homopolymeric RNAs. Error bars represent standard deviation of three replicates.

B   DIC micrographs of MBP-N in the presence of TEV (to cleave MBP from N to initiate LLPS) with indicated RNA. After cleavage of MBP, N phase separates. Apparent polyG RNA induces aggregation of MBP-N. Sample conditions: 50 µM MBP-N, 0.5 mg/ml RNA/polyX, 70 mM NaCl, 25°C, 50 mM Tris pH 7.4. Scale bar represents 80 µm.

and/or polyG gelation via formation of G-quadruplexes (Van Treeck *et al*, 2018). To determine if this increased and persistent turbidity of N in the presence of polyG RNA is indeed due to formation of arrested structures, rather than liquid forms, we imaged over time samples of MBP-N without or with TEV protease to cleave the MBP. Indeed, N in the presence of polyG RNA showed substantial irregularly shaped assemblies and no spherical, liquid-looking droplets (Fig 6B), similar to the irregular shaped structures caused by arrested phase separation previously observed for proline-arginine repeat peptides in the presence of polyG (Boeynaems *et al*, 2019). It is important to note that although buffer exchange via spin desalting columns should remove small RNA fragments, the size distribution of the homopolymeric RNAs are not the same and may impact the observed properties. Additionally, because of the complexity of both liquid and "gel-like" structures contributing to the turbidity timecourse which shows unusual non-monotonic behavior in some cases, we have chosen not to attempt to interpret these data quantitatively. In summary, these data suggest that the N-RNA interactions that enhance phase separation do not require particular sequences though polyG RNA stimulates formation of arrested structures that do not behave like liquids (Guillén-Boixet *et al*, 2020).

### N partitions into phase-separated forms of hnRNPA2, TDP-43, and FUS

SARS-CoV-2 N interacts with stress granule proteins (Gordon *et al*, 2020) and the interaction between one stress granule protein, hnRNPA1, and SARS-CoV N has been demonstrated (Luo *et al*, 2005). As we have previously shown co-partitioning of many granule-associated heterogenous nuclear ribonucleoproteins (hnRNPs) into liquid condensates including intermixing of FUS, hnRNPA2, and TDP-43 (Ryan *et al*, 2018; preprint: Ryan *et al*, 2020), we decided to test if full-length N could partition into liquid condensates formed by hnRNPA2, FUS, or TDP-43. We used conditions where N does not phase separate on its own (i.e., ~ 5 nM concentration with no RNA) to be sure that N was the "client" and the hnRNP was the "scaffold" protein, following common terminology to classify the molecules undergirding ("scaffold") or partitioning into ("clients") phase-separated droplets, respectively (Li *et al*, 2012; Alberti *et al*, 2019). We found that N partitions into hnRNPA2 LC and TDP-43 CTD droplets even when attached to the maltose binding protein (MBP) solubility tag (Fig 7A and B), though it did not partition into FUS LC droplets (Fig 7C). We note that this difference may have to do with unfavorability of partitioning MBP into droplets formed by FUS LC, which has an unusually low charged residue composition, not with details of FUS LC interactions with N. Indeed, N did partition into hnRNPA2 LC, TDP-43 CTD, and FUS LC droplets when N was cleaved from MBP (Fig 7C). We further tested whether N could partition into droplets formed by the full-length hnRNPs. We found that N was able to partition into full-length hnRNPA2, FUS, and TDP-43 droplets (Fig 7D–F). As the full-length hnRNPs are only able to undergo LLPS after cleavage of the MBP solubility tag, we could not attempt to observe partitioning of MBP-N and hnRNPs. These results indicate that N can enter liquid condensates formed by many human RNA-binding proteins, consistent with the presence of weak protein-protein interactions between N and human hnRNPs.

## Discussion

Virion assembly requires the formation of dense protein-nucleic acid compartments that sequester host cell proteins as a means of protection from the host immune system and concentrate viral components to increase the efficiency of replication (Novoa *et al*, 2005). Previous studies on herpes virus have reported the existence of perinuclear and cytoplasmic puncta in infected cells called prereplicative sites (Ishov & Maul, 1996; Uprichard & Knipe, 1997). Apart from SARS-CoV and SARS-CoV-2, a vast number of viruses like paramyxoviruses (Karlin *et al*, 2003), flaviviruses (Tompa & Csermely, 2004), and mononegaviruses such as rabies (Albertini *et al*, 2006; Nikolic *et al*, 2017), influenza A virus (Martín-Benito & Ortín, 2013; Turrell *et al*, 2013) and Lassa virus (Hastie *et al*, 2011) are also known to hijack their host cellular machinery via their highly disordered nucleoprotein with some of them resulting in the formation of cytoplasmic puncta such as the Negri bodies of *Mononegavirales* (Nikolic *et al*, 2017). Recently, the co-phase separation of full-length nucleoprotein (N) and phosphoprotein (P) of measles virus (MeV) was shown to require both the folded and the disordered domain of N, highlighting the importance of LLPS during MeV replication (Guseva *et al*, 2020). Understanding the mechanisms that underlie LLPS of SARS-CoV-2 N is essential to identify key potentially targetable steps in the viral replication cycle. To unravel the molecular details of this phenomenon, we tested the ability of N protein to undergo phase separation in the presence of RNA and other human ribonucleoproteins that are located in membraneless organelles of the eukaryotic cytoplasm and nucleoplasm (Ryan & Fawzi, 2019).

Here, we found that N of SARS-CoV-2 is able to phase separate and its phase behavior is tuned by pH, salt, and RNA concentration. At conditions chosen to resemble physiological pH and ionic strength, we observed the formation of *in vitro* N droplets after addition of TEV protease and RNA. Indeed, several reports point to SARS-CoV-2 N phase separation enhanced by RNA (Carlson *et al*, 2020; preprint: Cubuk *et al*, 2020; preprint: Iserman *et al*, 2020; preprint: Savastano *et al*, 2020). At lower pH, N phase separated without RNA, albeit to lower extents and the sphericity and hence the apparent fluidity were diminished (Appendix Fig S2), consistent with stabilization of protein-protein interactions. At lower salt conditions, N is also able to phase separate in the absence of RNA while at high salt phase separation is suppressed, consistent with electrostatic interactions that are screened at "physiological" ionic strength. Importantly, we find that addition of small amounts of RNA enhances phase separation, while higher RNA:protein ratios suppressed phase separation, suggesting that N of SARS-CoV-2 exhibits reentrant RNA-mediated phase separation, as observed for other RNA-binding proteins (Banerjee *et al*, 2017).

Several domains of N contribute to LLPS in the presence of RNA. Our observations that robust LLPS of N with RNA requires the prion-like disordered domains $N_{IDR}$ and linker$_{IDR}$ and the folded CTD dimerization domain are consistent with a model where multivalent contacts contribute to RNA-binding protein phase separation (Li *et al*, 2012). Furthermore, N phase separates at some (lower salt) conditions in the absence of RNA, even for the N variants containing domain deletions (Fig 5), suggesting that both protein–protein contacts and protein–RNA contacts contribute to LLPS. Removal of the $C_{IDR}$ in fact enhances LLPS, suggesting

that the $C_{IDR}$ lysine-rich stretch conserved from SARS-CoV may only serve as a secondary binding site for RNA (Luo *et al*, 2006) and the region may instead mediate key contacts with the membrane (M) protein as recently suggested (preprint: Lu *et al*, 2020). Therefore, current efforts aimed at probing the contacts formed by each domain and the structural features of the disordered domains will be key to understanding how N assembles with RNA (preprint: Cubuk *et al*, 2020).

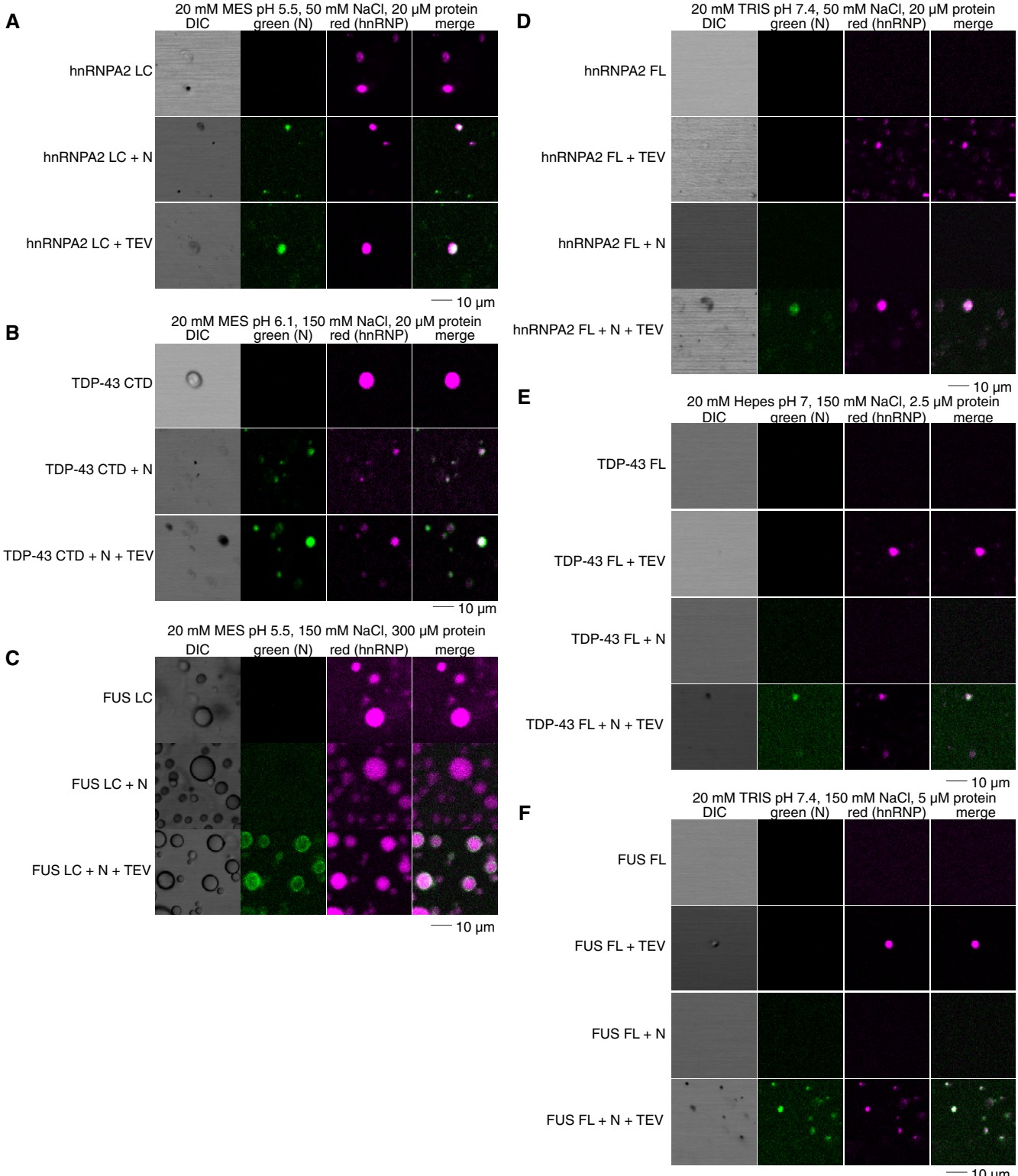

Figure 7.

**Figure 7. SARS-CoV-2 N partitions into liquid condensates of hnRNPs.**

A     N partitions into hnRNPA2 LC droplets even with the MBP tag attached.
B     N partitions into TDP-43 CTD droplets even with the MBP tag attached.
C     N partitions into FUS LC droplets only after cleavage of the MBP tag.
D–F   N partitions into hnRNPA2 FL (D), TDP-43 FL (E), FUS FL (F) droplets. Source data for panels (D, E, F) with N and TEV are available online for this figure.

Source data are available online for this figure.

We also compared the ability of homopolymeric low-complexity RNAs to induce LLPS of N. We found that LLPS does not require a mix of natural, structured RNAs as present in total yeast extracts. Indeed, all homopolymeric RNAs induced phase separation (Fig 6), consistent with other reports of distinct RNA sequences promoting SARS-CoV-2 N LLPS (Chen *et al*, 2020), but unlike LLPS of measles virus nucleocapsid particles that form after addition of a genomic RNA segment and polyA RNA but not polyU RNA (Milles *et al*, 2016). Here, we find that the assemblies of N and RNA formed predominantly spherical droplets with polyU, polyA, and polyC while N with polyG formed apparent fractal-like aggregate networks (Boeynaems *et al*, 2019). These structures of polyG with N strongly resemble the aggregates formed by proline-arginine dipeptide repeats in the presence of polyG, but not polyA, polyC, or polyU (Boeynaems *et al*, 2019), which again suggests that the unique stability of G-rich RNA structural elements result in these stable structures. Indeed, G-rich sequences can induce phase separation *in vitro* and in cells (Fay *et al*, 2017) and recent reports suggest that stable RNA secondary structural features play a key role in directing packaging of the viral genome (preprint: Iserman *et al*, 2020). We also note that a recent observation suggests that specific viral RNA sequences can enhance N phase separation, consistent with a picture where specific and non-specific RNA-protein interactions may cooperate in viral genome compaction (preprint: Iserman *et al*, 2020). Furthermore, that report also notes that distinct RNA sequences can alter material properties of N phase separation, which may be tuned for optimal genome packaging. Thus, here we demonstrate that the nucleocapsid protein from a virus of the *Coronaviridae* family undergoes phase separation with various RNAs *in vitro*, consistent with non-sequence-specific binding of N to RNA. Hence, phase separation of N with RNA may serve as a useful model for the process of genome packaging in SARS-CoV-2 and other coronaviruses.

Recent proteomic studies have constructed a putative SARS-CoV-2 protein interaction map where many RNA processing factors and stress granule regulation factors like G3BP1/2 (Yang *et al*, 2020) have been delineated as crucial nodes of the N interactome (Gordon *et al*, 2020). In addition, reports of molecules interacting with N of SARS-CoV highlight binding to granule-associated proteins—hnRNPA1 (Luo *et al*, 2005), the stress granule and phase-separating protein (Molliex *et al*, 2015); nucleophosmin (nucleolar phosphoprotein B23) (Zeng *et al*, 2008), the primary component of the phase-separated liquid granular component of the nucleolus (Feric *et al*, 2016), and cyclophilin A (Luo *et al*, 2004), an hnRNP chaperone (Pan *et al*, 2008). During the steps of genomic replication in the infected cells, the local concentration of cellular organelles is altered dramatically by the shuttling of the replication complexes in the cytoplasm and the anchoring of the structural proteins to the cellular membranes (Masters, 2006).

N may facilitate SARS-CoV-2 replication by recruiting stress granule components present in the host cellular environment (Cascarina & Ross, 2020) as suggested for other viruses (Monette *et al*, 2020). Here, we showed full-length N of SARS-CoV-2 can partition into liquid condensates composed of human ribonucleoproteins FUS, hnRNPA2 or TDP-43 full-length and their respective low-complexity domains. It is possible that abundant host cytoplasmic proteins, like these hnRNPs, serve as scaffolds to promote the formation of multicomponent N-RNA condensates to enable or accelerate viral replication. Given the potential role of phase separation in stabilizing nucleocapsid formation and the ability of viral nucleocapsid proteins to enter phase-separated assemblies formed by host cell RNA-binding proteins, it will be important to investigate if therapies targeting viral or host condensates could disrupt cycles of SARS-CoV-2 replication.

## Materials and Methods

### Constructs

- MBP-SARS-CoV-2 N full-length and domain deletions, soluble histag purification.
- hnRNPA2 LC, insoluble histag purification (Addgene: 98657).
- TDP-43 CTD, insoluble histag purification (Addgene: 98670)
- FUS LC, insoluble anion purification (Addgene: 98656).
- MBP-hnRNPA2 FL, soluble histag purification (Addgene: 139109).
- MBP-TDP-43 FL, soluble histag purification (Addgene: 104480).
- MBP-FUS FL, soluble histag purification (Addgene: 98651).

### MBP-N full-length and domain deletion expression and purification

MBP-tagged (pTHMT) full-length and domain deletion SARS-CoV-2 nucleocapsid proteins were expressed in *Escherichia coli* BL21 Star (DE3) cells (Life Technologies). Bacterial cultures were grown to an optical density of 0.7–0.9 before induction with 1 mM isopropyl-β-D-1-thiogalactopyranoside (IPTG) for 4 h at 37°C. Cell pellets were harvested by centrifugation and stored at −80°C. Cell pellets were resuspended in approximately 20 ml of 20 mM Tris 1M NaCl 10 mM imidazole pH 8.0 with one EDTA-free protease inhibitor tablet (Roche) for approximately 2 g cell pellet and lysed using an Emulsiflex C3 (Avestin). The lysate was cleared by centrifugation at 47,850 *g* for 50 min at 4°C, filtered using a 0.2 μm syringe filter, and loaded onto a HisTrap HP 5 ml column. The protein was eluted with a gradient from 10 to 300 mM imidazole in 20 mM Tris 1.0 M NaCl pH 8.0. Fractions containing MBP-N full-length or domain deletions were loaded onto a HiLoad 26/600 Superdex 200 pg column equilibrated in 20 mM Tris, 1.0 M NaCl, 1 mM EDTA

(added to further reduce protease degradation during processing), pH 8.0. Fractions with high purity were identified by SDS–PAGE and concentrated using a centrifugation filter with a 10 kDa cutoff (Amicon, Millipore).

### N full-length cleavage from MBP, MBP tag removal, and N gel filtration

MBP-N was incubated at ~ 600 μM in 20 mM Tris 1.0 M NaCl pH 8.0 with 0.03 mg/ml in-house TEV protease overnight. The protein was then buffer exchanged into 20 mM Tris 1.0 M NaCl pH 8.0 10 mM imidazole using a centrifugation filter with a 10 kDa cutoff (Amicon, Millipore). MBP (and hexahistidine tagged TEV) was then removed ("subtracted") using a HisTrap HP 5 ml column and flow through fractions containing cleaved N full-length were loaded onto a HiLoad 26/600 Superdex 200 pg column equilibrated in 20 mM Tris 1.0 M NaCl pH 8.0. Fractions from gel filtration were analyzed by SDS–PAGE.

### hnRNP purification

hnRNPA2 LC (Ryan *et al*, 2018), MBP-hnRNPA2 FL (preprint: Ryan *et al*, 2020), TDP-43 CTD (Conicella *et al*, 2016), MBP-TDP-43 FL (Wang *et al*, 2018), FUS LC (Burke *et al*, 2015), and MBP-FUS FL (Monahan *et al*, 2017) were purified as described.

### AlexaFluor labeling

Proteins were labeled with NHS-ester AlexaFluor dyes by diluting protein stocks into 20 mM HEPES pH 8.3 1 M NaCl (for N, hnRNPA2 FL, TDP-43 FL, and FUS FL) or 20 mM HEPES pH 8 with 8 M urea (FUS LC, hnRNPA2 LC, TDP-43 CTD). AlexaFluor dissolved in DMSO was added at less than 10% total reaction volume. Reactions were incubated for an hour and unreacted Alexa-Fluor was removed by desalting with 1 ml Zeba spin desalting columns equilibrated in the appropriate buffer for protein solubility. Labeled proteins were then concentrated and buffer exchanged into appropriate storage buffers and flash-frozen.

### Turbidity measurements

Turbidity was used to evaluate phase separation of 50 μM MBP-N full length in the presence of 0.01 mg/ml in-house TEV protease (~ 0.3 mg/ml in 50 mM Tris 1 mM EDTA 5 mM DTT pH 7.5 50% glycerol 0.1% Triton-X-100) in the appropriate conditions. To test the effect of pH on LLPS, the experiment was conducted in 50 mM Tris 183 mM NaCl pH 7.4, 20 mM MES 183 mM pH 6.1, 20 mM MES 183 mM pH 5.5, 20 mM MES 183 mM pH 4.9, 20 mM MES 183 mM pH 4.5 with 0.3 mg/ml desalted (into the appropriate buffer using a Zeba 0.5 ml spin column) torula yeast RNA extract in the appropriate buffer conditions. To test the effect of different salt concentrations on LLPS, the experiments were conducted in 50 mM Tris pH 7.4 with 60, 100, 300, or 1,000 mM NaCl with 1.1 mg/ml desalted torula yeast RNA extract. To test the effect of RNA on LLPS, the experiments were conducted in 50 mM Tris 100 mM NaCl pH 7.4 with 0, 1.1 or 2.3 mg/ml desalted torula yeast RNA extract. To test the effect of domain deletions and homopolymeric RNAs (polyA, polyC, polyG, polyU) on LLPS, the experiments were

conducted in 50 mM Tris 70 mM NaCl pH 7.4 with 0.5 mg/ml desalted torula yeast RNA or desalted polyX RNA (Sigma). Turbidity experiments were performed in a 96-well clear plate (Costar) with 70 μL samples sealed with optical adhesive film to prevent evaporation (MicroAmp, Thermo Fisher). The absorbance at 600 nm was monitored over time using a Cytation 5 Cell Imaging Multi-Mode Reader (BioTek) at 5 min time intervals for up to 12 h with mixing and subtracted from a blank with no turbidity (e.g., buffer). Experiments were conducted in triplicate and averaged.

### DIC microscopy

For 50 μM MBP-N full-length and domain deletions, the samples were incubated with 0.03 mg/ml in-house TEV protease for ~ 20 min before visualization. Samples were spotted onto a glass coverslip and droplet formation was evaluated by imaging with differential interference contrast on an Axiovert 200M microscopy (Zeiss).

### hnRNP mixing microscopy

Each protein was prepared for microscopy based on our previously established methods for that protein (Burke *et al*, 2015; Conicella *et al*, 2016; Monahan *et al*, 2017; Ryan *et al*, 2018; Wang *et al*, 2018; preprint: Ryan *et al*, 2020). Briefly, hnRNPA2 was diluted from 8 M urea into the appropriate buffer to a final concentration of 150 mM urea and appropriate protein concentration; hnRNPA2 FL with a C-terminal MBP tag was diluted from 1 M NaCl to 50 mM NaCl and appropriate protein concentration; TDP-43 CTD was desalted into MES pH 6.1 using a 0.5 ml Zeba spin desalting column and diluted to the appropriate protein concentration; TDP-43 FL with a C-terminal MBP tag was diluted from storage buffer into appropriate buffer at indicated concentration; FUS LC was diluted from 20 mM CAPS to appropriate concentration in indicated buffer; and FUS FL with an N-terminal MBP tag was diluted from 1 M NaCl to a final NaCl concentration of 150 mM and indicated protein concentration. 1 μl of 0.3 mg/ml TEV was added as appropriate, if no TEV was needed for the sample, TEV storage buffer was added instead. Buffer conditions for each protein are listed below:

- hnRNPA2 LC: 20 μM hnRNPA2 LC, 20 mM MES pH 5.5, 50 mM NaCl, 150 mM urea (residual), ~ 5 nM AlexaFluor labeled protein (each, if both N and hnRNP are present).
- hnRNPA2 FL: 20 μm hnRNPA2 FL, 20 mM TRIS pH 7.4, 50 mM NaCl, ~ 5 nM AlexaFluor labeled protein (each, if both N and hnRNP are present).
- TDP-43 CTD: 20 μM TDP-43 CTD, 20 mM MES pH 6.1, 150 mM NaCl, ~ 5 nM AlexaFluor labeled protein (each, if both N and hnRNP are present).
- TDP-43 FL: 2.5 μM TDP-43 FL, 20 mM HEPES pH 7, 150 mM NaCl, 1 mM DTT, ~ 5 nM AlexaFluor labeled protein (each, if both N and hnRNP are present).
- FUS LC: 300 μM FUS LC, 20 mM MES pH 5.5, 150 mM NaCl, ~ 5 nM AlexaFluor labeled protein (each, if both N and hnRNP are present).
- FUS FL: 5 μM FUS FL, 20 mM TRIS pH 7.4, 150 mM NaCl, ~ 5 nM AlexaFluor labeled protein (each, if both N and hnRNP are present).

Fluorescence confocal microscopy images were taken on an LSM 880 (Zeiss). AlexaFluor-tagged proteins were doped in at 0.2 μl (~ 5 nM final concentration) to prevent oversaturation of the detector. Snapshots were taken of the red, green, and brightfield channels and merged using ImageJ (NIH).

## Data availability

This study includes no data deposited in external repositories. Datasets produced in this study are available by request to the corresponding author. Plasmids generated herein can be found at https://www.addgene.org/Nicolas_Fawzi/ for public distribution.

**Expanded View** for this article is available online.

### Acknowledgements

We thank Gerwald Jogl and Walter Atwood for helpful input and Geoff Williams and Christoph Schorl for technical assistance. We thank Abigail Janke and Alexander Conicella for making available stocks of their FUS and TDP-43 full-length protein, respectively. Research was supported by a COVID-19 Research Seed Award from Brown University (to M.T.N., N.L.F, Gerwald Jogl, and Walter Atwood), the Division of Biology and Medicine at Brown University, the National Institute of General Medical Sciences R01GM118530 (to N.L.F.), the National Institute of Neurological Diseases and Stroke and the National Institute on Aging R01NS116176 (to N.L.F.). A.C.M. was supported by an NSF graduate fellowship (1644760). V.H.R. was supported by the National Institutes of Health F31NS110301.

### Author contributions

TMP performed bioinformatics analysis and experiments on N domain deletions. ACM performed phase separation turbidity assays and microscopy of N. VHR performed N partitioning assays and turbidity screening in the presence of divalent metal salts. TMP and VHR performed the assays using homopolymeric RNAs. SW designed the expression construct and purified protein. NLF and MTN contributed to research design and funding acquisition. TMP led the writing of the manuscript with text, figures, and comments provided by all authors.

### Conflict of interest

N.L.F. is a member of the scientific advisory board of Dewpoint Therapeutics. The authors declare no other conflicts of interest.

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
