## [Review Process File · The EMBO Journal]

SARS-CoV-2 nucleocapsid protein phase-separates with RNA and with human hnRNPs

Theodora Myrto Perdikari, Anastasia C. Murthy, Veronica H. Ryan, Scott Watters, Mandar T. Naik, and Nicolas L. Fawzi

DOI: 10.15252/embj.2020106478

Corresponding author(s): Nicolas Fawzi (nicolas_fawzi@brown.edu)

Review Timeline:

Submission Date:	7th Aug 20
Editorial Decision:	17th Aug 20
Revision Received:	14th Nov 20
Accepted:	16th Nov 20

Editor: Karin Dumstrei

Transaction Report:

Dear Nick,

Thanks for submitting your manuscript to The EMBO Journal along with your point-by-point response to the referee comments from a previous journal. Your study has now been seen by one referee who also had access to your point-by-point response.

As you can see below, the referee appreciate the findings reported and also find the analysis technical strong. The referee has some specific comments that should be straight forward enough to address without much further work. The referee also doesn't agree with the comments raised by one of the previous referees that it could be predicted by the presence of IDR domains that N protein would undergo phase separation. The referee also finds that the technical issues raised by the previous referees have been adequately addressed. Given this positive assessment, we are interested in publishing your paper here.

One thing that I would like to discuss with you is that the referee suggests a few additional experiments to make the analysis more complete (If revision experiments...). I think they are good suggestions - do you have any data on hand to address them or what would it take to get some insight into the proposed experiments. Let's discuss this further.

Thank you for the opportunity to consider your work for publication. I look forward to your revision.

with best wishes

Karin

Karin Dumstrei, PhD
Senior Editor
The EMBO Journal

When assembling figures, please refer to our figure preparation guideline in order to ensure proper formatting and readability in print as well as on screen:
<http://bit.ly/EMBOPressFigurePreparationGuideline>

IMPORTANT: When you send the revision we will require
- a point-by-point response to the referees' comments, with a detailed description of the changes

made (as a word file).

- a word file of the manuscript text.

- individual production quality figure files (one file per figure)

- a complete author checklist, which you can download from our author guidelines (<https://www.embopress.org/page/journal/14602075/authorguide>).

- Expanded View files (replacing Supplementary Information)

Further information is available in our Guide For Authors:

The revision must be submitted online within 90 days; please click on the link below to submit the revision online before 15th Nov 2020.

Referee #1:

Major comment on basis of the current manuscript:

. The image quality of several microscopy images needs to be improved (magnification, DIC alignment; see below)

. Figure 3B: I appreciate the authors want to provide a large field of view to represent the effect of RNA on N-protein phase separation. However, as they draw conclusions regarding the material properties purely based on condensate morphology, this is nearly impossible to judge from the small magnification. The authors should either provide a high magnification image or an inset with higher zoom in existing images to support this conclusion. As the manuscript lacks any kind of detailed studies regarding material properties (droplet fusion events or lack of those, FRAP), the basis for those claims must be absolutely clear for the reader!

. Figure 5B: Same criticism as for Figure 3B. The authors claim to have improved the data presentation, but this does not appear to be the case here. I wonder whether the DIC alignment was well performed, the images certainly drop in quality to previous DIC images of N-protein condensates in Figure 3 and 4! Also here, a higher magnification might also help to visualize more clearly whether N proteins forms liquid-like droplets or rather amorphous condensates. As the authors do not discuss the "-Tev" condition (e.g. as to why are only transient condensate formed in presence of polyG?), those panels can be removed to help to transmit the point the authors want to make.

. Figure 6: I find the level of phase separation observed for the FL RBPs surprisingly low, in some panels, the authors merely show one or very few droplets to illustrate the partitioning phenomenon of the N-protein. Can the authors exclude the fluorophore label is interfering with RBP LLPS (even though only doped in)? Images displaying more condensates would be a more convincing. Have the authors found any protein condensates the N proteins DOES NOT partition into (suggesting some specificity)? I am not too convinced by the lack of partitioning of MBP-N into FUS LC droplets here - this might simply due to interference of the MBP tag. Is there any kind of promoting or concentration effect? Either the N protein promoting phase separation of RBPs (or their LCDs) or vice versa?

. The authors interpret their findings that all kind of homopolymeric RNA promote N protein phase transition as lack of sequence specificity. They do however refer to other studies (preprint) demonstrating RNA specificity (using viral RNA). The observation that polyG causes amorphous protein aggregates, while for example polyU/N protein appears more liquid like, resembles observations by Boeynaems et al (2018; PNAS) and could be indicative of a role for RNA structure in this process. This could be addressed by for example performing those kind of experiments using RNA sequences forming strong secondary structures (hairpins, G-quadruplex) but at least should be appropriately discussed! In the end, RNA secondary structure could well be a packing signal of the viral RNA.

Minor comments:

. Figure 1 is not particularly illustrative or meaningful for the reader. Instead of displaying the amino acid composition of the three IDRs, an order/disorder plot will be more informative. If the amino acid composition charts are to be included, a comparison to an ordered domain (NTD and/or CTD) would illustrate better if the amino acid composition is of rather low complexity or biased towards specific amino acids.

. Figure 3A: please indicate salt concentration used in the figure legend.

. Please provide reference for the statement that bivalent cations modulate RNP complex behaviour (p 8)- right now the impact of this statement for N phase transition is not clear.

. Please indicate concentration (as fold) for protease inhibitors in the protein purification method section, "1 tablet" is not meaningful without knowledge of the buffer volume

If revision experiments are feasible:

- Can the author provide insight into which IDR of the N protein is involved in phase separation with either RNA or RBPs? Are all required or is one the main driver? Does their position or rather amino acid composition have the larger impact?
- Use of RNA with specific secondary structures in N protein condensation could provide insight into the packing mechanism of viral RNA (as it is rather unlikely any kind of RNA will be packed into

virions).

- If possible, provide experiments directly addressing material properties of N protein condensates.

Referee #1:

Major comment on basis of the current manuscript:

. The image quality of several microscopy images needs to be improved (magnification, DIC alignment; see below)

We appreciate the reviewer's comments regarding quality of the images. We have addressed these comments below

. Figure 3B: I appreciate the authors want to provide a large field of view to represent the effect of RNA on N-protein phase separation. However, as they draw conclusions regarding the material properties purely based on condensate morphology, this is nearly impossible to judge from the small magnification. The authors should either provide a high magnification image or an inset with higher zoom in existing images to support this conclusion. As the manuscript lacks any kind of detailed studies regarding material properties (droplet fusion events or lack of those, FRAP), the basis for those claims must be absolutely clear for the reader!

We have now increased the size of these images to support the conclusions as the reviewer has suggested. We also agree with the reviewer that we have not thoroughly investigated the material states of these assemblies. We have reworked language throughout to avoid making claims about details of material states, which are not the point of this manuscript.

. Figure 5B: Same criticism as for Figure 3B. The authors claim to have improved the data presentation, but this does not appear to be the case here. I wonder whether the DIC alignment was well performed, the images certainly drop in quality to previous DIC images of N-protein condensates in Figure 3 and 4! Also here, a higher magnification might also help to visualize more clearly whether N proteins forms liquid-like droplets or rather amorphous condensates. As the authors do not discuss the "-Tev" condition (e.g. as to why are only transient condensate formed in presence of polyG?), those panels can be removed to help to transmit the point the authors want to make.

We agree that DIC alignment may be insufficient for optimal clarity in the original images, however, we believe that with droplets that are larger than these repeated artifacts, the scientific value of the original images are not in doubt. Nevertheless, we agree with the reviewer and for optimal clarity, we have repeated the experiment to correct the issues with the images (results are effectively the same) and have also removed the -TEV condition to focus on the main point as suggested by the reviewer – this also allows us to make the images bigger. Finally, we have removed analysis of the shape of the condensates as providing information on the material state, except for polyG, which has a distinctly different morphology.

. Figure 6: I find the level of phase separation observed for the FL RBPs surprisingly

low, in some panels, the authors merely show one or very few droplets to illustrate the partitioning phenomenon of the N-protein. Can the authors exclude the fluorophore label is interfering with RBP LLPS (even though only doped in)? Images displaying more condensates would be a more convincing. Have the authors found any protein condensates the N proteins DOES NOT partition into (suggesting some specificity)? I am not too convinced by the lack of partitioning of MBP-N into FUS LC droplets here - this might simply due to interference of the MBP tag. Is there any kind of promoting or concentration effect? Either the N protein promoting phase separation of RBPs (or their LCDs) or vice versa?

We appreciate the reviewer's concerns. Our experiments do not use polymeric crowding agents to induce LLPS which we believe may have more impacts on droplets beyond simple shifting the phase diagram. Therefore, our dilute concentrations result in few droplets. Hence, we have no reason to be concerned that the small amount of fluorescent protein inhibits LLPS and we have performed these experiments in the past examining each RBP phase separation (Ryan et al Mol Cell 2018, Burke et al Mol Cell 2015, Conicella et al Structure 2016) and have also used this approach to evaluate partitioning (Ryan et al biorxiv 2020). We have added source data of these images with larger fields of view with multiple small droplets that interested readers can examine with high magnification.

We appreciate the reviewer's insight and we do agree that the lack of partitioning of MBP-N into FUS LC droplets may have more to do with MBP interactions with (or difficultly partitioning into) FUS LC than with N interactions with FUS LC, so we have clarified comments suggesting a FUS LC / N difference to explicitly make this clear. It is possible that N can promote phase separation of RBPs but it is not trivial to perform these experiments (indeed we are still working on developing and optimizing a precise approach to do this) and so we have not attempted to ask this question here.

The revised text is as follows:

*We found that N partitions into hnRNPA2 LC and TDP-43 CTD droplets even when attached to the maltose binding protein (MBP) solubility tag (**Figure 7A-B**), though it did not partition into FUS LC droplets (**Figure 7C**). We note that this difference may have to do with unfavorability of partitioning MBP into the unusually low charged residue sequence of FUS LC. Indeed, N did partition into hnRNPA2 LC, TDP-43 CTD, and FUS LC droplets when N was cleaved from MBP (**Figure 7C**).*

. The authors interpret their findings that all kind of homopolymeric RNA promote N protein phase transition as lack of sequence specificity. They do however refer to other studies (preprint) demonstrating RNA specificity (using viral RNA). The observation that polyG causes amorphous protein aggregates, while for example polyU/N protein appears more liquid like, resembles observations by Boeynaems et al (2018; PNAS) and could be indicative of a role for RNA structure in this process. This could be addressed by for example performing those kind of experiments using RNA sequences forming strong secondary structures (hairpins, G-quadruplex) but at least should be appropriately discussed! In the end, RNA secondary structure could well be a packing signal of the viral RNA.

We agree with the reviewer that experiments with specific viral RNAs would be interesting. We feel it is beyond the scope of this manuscript and we refer here to the preprints covering this aspect. We have now expanded the text to appropriately discuss the work of Boeynaems et al 2018 and others both in results and discussion.

Minor comments:

. Figure 1 is not particularly illustrative or meaningful for the reader. Instead of displaying the amino acid composition of the three IDRs, an order/disorder plot will be more informative. If the amino acid composition charts are to be included, a comparison to an ordered domain (NTD and/or CTD) would illustrate better if the amino acid composition is of rather low complexity or biased towards specific amino acids.
We have changed figure one to a domain diagram with disorder/order.

. Figure 3A: please indicate salt concentration used in the figure legend.
We have now added the salt concentration and clarified the legend.

. Please provide reference for the statement that bivalent cations modulate RNP complex behaviour (p 8)- right now the impact of this statement for N phase transition is not clear.
We have added references and clarified this statement.

. Please indicate concentration (as fold) for protease inhibitors in the protein purification method section, "1 tablet" is not meaningful without knowledge of the buffer volume
We have added the buffer volume as well as the estimated cell pellet mass, both of which are important factors.

If revision experiments are feasible:

- Can the author provide insight into which IDR of the N protein is involved in phase separation with either RNA or RBPs? Are all required or is one the main driver? Does their position or rather amino acid composition have the larger impact?

We have now added new experiments addressing some of this information in new Figure 5. We find that indeed some IDRs and folded domains have a large impact on LLPS while others do not, providing insight into the relative role of the domains in multivalent interactions. We agree it would be interesting to observe if alter position also impacts their role, but we have not been able to address this yet as moving domains is significantly more challenging to construct and creates an explosion of possibilities to test.

- Use of RNA with specific secondary structures in N protein condensation could provide insight into the packing mechanism of viral RNA (as it is rather unlikely any kind of RNA will be packed into virions).

As we mentioned above, we have not been able to add this data within this timeline and scope, though we agree that these are important experiments. Given that the ratio of N to viral RNA is very high, it is not likely to be a unique sequence of RNA that binds N and we believe sequence-independent interaction contribute to packing, as in histone proteins.

- If possible, provide experiments directly addressing material properties of N protein condensates.

We appreciate the potential additional insight this would add but have elected not to address this directly. In our experience, interactions with the slide surface can limit the usefulness of performing FRAP experiments attempting to quantitatively distinguish different phases and can change rapidly (faster than the experiment) yet the morphology provides an important indicator of the material state at the time the structure was formed (e.g. round due to liquid of “low” viscosity while irregular indicates incomplete liquidity or very high viscosity – truly distinguishing these two options for irregular structures is difficult by FRAP). As the topic of the manuscript is not on the material states, we have removed much of the conclusions highlighting change in morphology as a proxy for material state.

Dear Nick,

Thank you for submitting your revised manuscript to The EMBO Journal. I have now had a chance to take a careful look at everything and I appreciate the introduced changes. I am therefore very pleased to accept the manuscript for publication here.

With best wishes

Karin

Karin Dumstrei, PhD
Senior Editor
The EMBO Journal

Please note that it is EMBO Journal policy for the transcript of the editorial process (containing referee reports and your response letter) to be published as an online supplement to each paper. If you do NOT want this, you will need to inform the Editorial Office via email immediately. More information is available here: https://emboj.embopress.org/about#Transparent_Process

Your manuscript will be processed for publication in the journal by EMBO Press. Manuscripts in the PDF and electronic editions of The EMBO Journal will be copy edited, and you will be provided with page proofs prior to publication. Please note that supplementary information is not included in the proofs.

Should you be planning a Press Release on your article, please get in contact with embojournal@wiley.com as early as possible, in order to coordinate publication and release dates.

If you have any questions, please do not hesitate to call or email the Editorial Office. Thank you for your contribution to The EMBO Journal.

Corresponding Author Name: Nicolas L. Fawzi

Manuscript Number: EMBOJ-2020-106478R